# Nephrotic Syndrome and Statin Therapy: An Outcome Analysis

**DOI:** 10.3390/medicina59030512

**Published:** 2023-03-06

**Authors:** Ruxandra Busuioc, Gabriel Ștefan, Simona Stancu, Adrian Zugravu, Gabriel Mircescu

**Affiliations:** 1Nephrology Department, University of Medicine and Pharmacy “Carol Davila”, 050474 Bucharest, Romania; 2Nephrology Department, “Dr Carol Davila” Teaching Hospital of Nephrology, 010731 Bucharest, Romania

**Keywords:** statin therapy, nephrotic syndrome, remission rate, end stage kidney disease, major cardiovascular events, thrombotic complications

## Abstract

*Background and Objectives*: Hypercholesterolemia in patients with nephrotic syndrome (NS) may predispose to cardiovascular events and alter kidney function. We aimed to evaluate statins efficiency in NS patients under immunosuppression using four endpoints: remission rate (RR), end-stage kidney disease (ESKD), major cardiovascular events (MACE), and thrombotic complications (VTE). *Materials and Methods*: We retrospectively examined the outcome at 24 months after diagnosis of 154 NS patients (age 53 (39–64) years, 64% male, estimated glomerular filtration rate (eGFR) 61.9 (45.2–81.0) mL/min). During the follow-up, the lipid profile was evaluated at 6 months and at 1 and 2 years. *Results*: The median cholesterol level was 319 mg/dL, and 83% of the patients received statins. Patients without statins (17%) had similar age, body mass index, comorbidities, blood lipids levels, NS severity, and kidney function. The most used statin was simvastatin (41%), followed by rosuvastatin (32%) and atorvastatin (27%). Overall, 79% of the patients reached a form of remission, 5% reached ESKD, 8% suffered MACE, and 11% had VTE. The mean time to VTE was longer in the statin group (22.6 (95%CI 21.7, 23.6) versus 20.0 (95%CI 16.5, 23.5) months, *p* 0.02). In multivariate analysis, statin therapy was not associated with better RR, kidney survival, or fewer MACE; however, the rate of VTE was lower in patients on statins (HR 2.83 (95%CI 1.02, 7.84)). *Conclusions*: Statins did not improve the remission rate and did not reduce the risk of MACE or ESKD in non-diabetic nephrotic patients. However, statins seemed to reduce the risk of VTE. Further randomized controlled studies are needed to establish statins’ role in NS management.

## 1. Introduction

Abnormal lipid metabolism is common in patients with nephrotic syndrome, where hypercholesterolemia and hypertriglyceridemia are seen in 90 and 78% of cases, respectively [1,2]. The underlying mechanisms of hyperlipidemia in nephrotic syndrome are only partially understood and involve both enhanced synthesis and decreased metabolism of lipoproteins. Thus, apoB lipoproteins (low-density lipoproteins, LDL) are primarily increased, and their content is higher in cholesterol, triglycerides, and phospholipids than in the general population, whereas high-density lipoproteins (HDL), although usually normal or variable, are dysfunctional [2,3,4]. Persistent hypercholesterolemia, low blood volume, and hypercoagulability in the setting of nephrotic syndrome predispose patients to cardiovascular events and may alter kidney outcomes [5,6].

Besides the specific immunosuppressive therapy for the glomerular disease, the management of hyperlipidemia in patients with nephrotic syndrome follows the guidelines recommended for the general population and use the same lipid-lowering agents. Statins are well-tolerated and effective in correcting, at least partially, the abnormal lipid profile in patients with nephrotic syndrome [7]. However, evidence for the presumed cardiovascular event reduction or kidney disease improvement due to statin therapy is lacking in patients with nephrotic syndrome [7,8,9].

Therefore, we aimed to evaluate statins efficiency in patients with nephrotic syndrome under immunosuppressive therapy using four hard endpoints: remission rate, end-stage kidney disease, major cardiovascular events, and thrombotic complications.

## 2. Materials and Methods

### 2.1. Patients and Study Design

This retrospective unicentric study included consecutive non-diabetic adult patients newly diagnosed with nephrotic syndrome (24 h proteinuria >3.5 g and hypoalbuminemia <3.5 g/dl) between 1 January 2010 and 31 December 2015 at “Dr. Carol Davila” Teaching Hospital of Nephrology.

The patients were followed from the time of kidney biopsy to the following four distinct outcomes or until 24 months after diagnosis:*kidney replacement therapy initiation* (dialysis initiation or kidney transplantation).*major cardiovascular event* (MACE) defined as cardiovascular death, myocardial infarction, or ischemic stroke.*thrombotic complications* including deep vein thrombosis, renal vein thrombosis, and pulmonary embolism.*complete remission* defined as proteinuria under 0.5 g per 24 h, serum albumin of at least 3.5 g per deciliter, and stable eGFR (eGFR remaining unchanged or declining by <15% during follow-up).

During the follow-up, the lipid profile was evaluated at 6 months and a year and 2 years after the kidney biopsy.

Patients were excluded when they were younger than 18 years at the time of the kidney biopsy, if they had diabetes mellitus or cancer, if they were untreated with immunosuppressants, or if the follow-up was less than 24 months.

### 2.2. Covariates (Measurements) and Treatment

Electronic medical records were reviewed for demographics, presentation characteristics, thrombotic complications, outcome data, and laboratory parameters (i.e., serum creatinine, proteinuria, serum albumin, blood lipids, and inflammation). The Charlson comorbidity score (available at: https://www.mdcalc.com/calc/3917/charlson-comorbidity-index-cci; URL accessed on 10 June 2021) was used to assess the burden of comorbidities in the studied patients. It is based on a weighted sum of the presence and severity of 17 various conditions. The averages of the serum albumin and urinary protein values in 6 month periods are presented as the time-averaged serum albumin and proteinuria.

Only patients receiving immunosuppressive therapy were included in the analyses. The choice of the immunosuppressive treatment was at the discretion of the nephrologist in charge.

### 2.3. Statistical Analysis

Descriptive statistics were summarized as mean ± SD or median (quartile 1, quartile 4) for continuous variables, and frequency distribution is presented as percentages for categorical variables. Group comparisons were performed with Student’s *t*-test, χ^2^ test, and Mann–Whitney U test, as appropriate.

Survival analyses were conducted with the Kaplan–Meier method, and the log rank test was used for comparisons. Multivariate Cox proportional hazard analyses were performed to identify independent predictors of the studied endpoints. The results were expressed as hazard ratio (HR) and 95% confidence interval (CI).

All statistical tests were two-sided, and a *p* < 0.05 was considered significant. Statistical analyses were performed using the SPSS program (SPSS version 26, Chicago, IL, USA).

### 2.4. Ethics Approval

The study was conducted with the provisions of the Declaration of Helsinki, and the protocol was approved by the local ethics committee (“Dr. Carol Davila” Teaching Hospital of Nephrology, Bucharest, Romania, approval number 2021-012, 3 March 2021). Since all data were anonymized, informed consent was not obtained from individual patients.

## 3. Results

We found 213 patients with nephrotic syndrome in our databases during the study period. Based on the exclusion criteria, 59 cases were excluded; the main reasons were: incomplete follow-up (30), childhood onset of the nephrotic syndrome, (5) and missing data on statin therapy (24).

The final cohort comprised 154 patients, with a median age of 53 years at the time of the kidney biopsy and with a male sex predominance (64%). The most frequent cause of nephrotic syndrome was membranous nephropathy (55%), followed by minimal change disease (31%), focal and segmental glomerulosclerosis (11%), and membranoproliferative glomerulonephritis (3%). All patients received an immunosuppression therapy based on corticotherapy, i.e., 26% received corticotherapy only, 58% corticotherapy plus cyclophosphamide, and 16% corticotherapy plus cyclosporine. There were no differences between the two studied groups in regard to the type of immunosuppressive drugs (Table 1).

The baseline characteristics of the included patients are presented in Table 1. The median eGFR was 70 mL/min at diagnosis, and all included patients had full-blown nephrotic syndrome (i.e., median proteinuria, 6.4 g/g and hypoalbuminemia, 2.9 g/dL) at presentation. Almost half of the patients had arterial hypertension, but the comorbidity burden evaluated with the Charlson index was rather mild with a median score of one.

The median cholesterol level was 319 mg/dL, and 83% of the patients received statin therapy. However, the patients without statin therapy had similar age, body mass index, comorbidities, blood lipids levels (cholesterol, triglycerides, and total lipids), nephrotic syndrome severity, and kidney function (Table 1).

The most frequent prescribed statin was simvastatin (41%), followed by rosuvastatin (32%) and atorvastatin (27%). The median dose of the prescribed statins was 20 mg; there were no differences in dosing between the statin categories.

During the follow-up period, we found no differences in the cholesterol levels, neither depending on the used statins nor between statin-treated and untreated patients, at baseline, 6 months, and 1 and 2 years (Figure 1).

Overall, 79% of the studied patients reached a form of remission (Table 1). Complete remission rates were 43% after 6 months, 55% after 12 months, and 65% after 24 months.

The mean time to cumulative remission was 13.7 (95%CI 12.2, 15.2) months. There was no relationship between statin therapy and time to remission (statin, 13.9 (95%CI 12.4, 15.6), no statin, 12.6 (95%CI 8.8, 16.3) months, *p* 0.4) (Figure 2A).

Eight patients (5%) started kidney replacement therapy (Table 1). The primary renal diseases were focal and segmental glomerulosclerosis (*n* = 3, 37.5%), membranous nephropathy (*n* = 3, 37.5%), and minimal change disease (*n* = 2, 25%). Kidney survival at 6, 12, and 24 months was 97, 96, and 93%, respectively. The mean kidney survival time was similar between the two studied groups (statin, 23.4 (95%CI 22.8, 24.0), no statin, 22.4 (95%CI 20.3, 24.0) months, *p* 0.5) (Figure 2B). Moreover, there were no differences in kidney survival depending on the type of statin used (Kaplan–Meier analysis, log rank test, *p* 0.1).

Twelve (8%) patients suffered from a major adverse cardiovascular event; however, no death was registered during the follow-up period (Table 1). The mean time to MACE was 23.2 (95%CI 22.7, 23.8) months. We report no relationship between treatment with statin and time to MACE (statin, 23.1 (95%CI 22.4, 23.8), no statin, 24.0 (95%CI 24.0, 24.0) months, *p* 0.6) (Figure 2C).

Thrombotic complications were present only in 11% of the patients and included pulmonary embolism, renal vein thrombosis, and deep vein thrombosis. Interestingly, the patients who were not on statin therapy had a significantly higher percent of thrombotic complications during the follow-up period (23 vs. 9%, *p* 0.03) (Table 1). Furthermore, the mean time to a thrombotic event was significantly shorter in these patients (no statin, 20.0 (95%CI 16.5, 23.5), statin, 22.6 (95%CI 21.7, 23.6) months, *p* 0.02) (Figure 2D).

In multivariate analysis, statin therapy was not associated with better remission rate, kidney survival, or fewer MACE; however, the rate of thrombotic complications was significantly lower in patients who received statins (Table 2). In addition, the protective effect of statins (HR 3.16 (95%CI 1.13, 8.83)) for thrombotic complications remained after adjusting for time-averaged albumin (HR 0.85 (95%CI 0.34, 2.12)) and proteinuria (HR 1.05 (95%CI 0.92, 1.21)).

Serious side effects were not reported in the studied patients. Only two patients had a more than three times upper the normal limit elevation in alanine aminotransferase levels, and one patient complained of myalgias; in all these cases, statin therapy was stopped.

## 4. Discussion

In this retrospective study involving nephrotic patients treated with immunosuppressives, statin therapy did not improve the cholesterol levels. Moreover, there were no differences in nephrotic syndrome remission rates, in the proportion of patients who initiated kidney replacement therapy, and in the rate of major cardiovascular events during the two-year follow-up. However, patients on statins had fewer thrombotic complications.

The particularity of our research is that we studied—for the first time to our knowledge—the relationship between statins and these four hard endpoints in non-diabetic patients with nephrotic syndrome who received immunosuppressive therapy. Currently, statins are the norm in the management of hyperlipidemia of nephrotic syndrome, despite the paucity of data to prove their efficiency in this setting [7,8,9,10,11].

Statins seem to be well tolerated and effective in correcting, at least partially, the abnormal lipid profile in patients with nephrotic syndrome [9,10]. However, the benefits and ability of statins to slow the progression of chronic kidney disease remain unproven and largely controversial [12]. Thomas et al. performed a randomized controlled trial on thirty adult patients with nephrotic syndrome or significant proteinuria (>1 g/day) who were randomized to simvastatin or placebo therapy and found no significant differences between the groups in proteinuria levels, rise in serum creatinine, or decline in plasma inulin clearance [8]. In line with this, we report no relationship between statin therapy and remission rate or kidney replacement therapy initiation in our cohort.

These results might challenge the dogma that statins are efficient in hyperlipidemia management in patients with nephrotic syndrome treated with immunosuppressive drugs. Statin effect on the cholesterol level and on MACE could be confounded by the immunosuppressive treatment which leads to the remission of the nephrotic syndrome.

Immunosuppressant-mediated hypercholesterolemia has also been reported, especially for cyclosporine [13]. However, the mechanisms behind immunosuppressant-mediated hypercholesterolemia are not completely understood. In the case of cyclosporine, it appears that many steps of lipid metabolism can be disturbed: lipoprotein synthesis, lipolysis, uptake, and clearance [13]. In our population, all patients received immunosuppressive treatment, and only 16% of them were on a cyclosporine regimen. Moreover, there were no differences between the two studied groups concerning the immunosuppression agents. Therefore, the confounding risk of immunosuppressant-mediated hypercholesterolemia in our study was reduced.

There are numerous epidemiologic studies and randomized clinical trials that have established hypercholesterolemia as a major risk factor in the pathogenesis of atherosclerotic cardiovascular disease. Nevertheless, there are few studies to confirm these findings in patients with nephrotic syndrome.

Ordonez et al. compared 142 non-diabetic adult patients with nephrotic syndrome with matched controls, and those with nephrotic syndrome had a higher risk of myocardial infarction (relative risk [RR], 5.5, 95%CI 1.6–18.3) but a non-significantly higher risk of coronary death (RR, 2.8, 95%CI 0.7–11.3) [5]. In addition, Dogra et al. showed that statin therapy improved brachial artery endothelial function measured by flow-mediated dilatation in patients with nephrotic syndrome [14]. Nevertheless, persistent nephrotic syndrome and hyperlipidemia seem to be risk factors for atherosclerotic cardiovascular disease, especially if other cardiovascular risk factors are present [15]. However, the low incidence of MACE in our cohort could be explained by the rather mild Charlson score and the relatively short follow-up time for a population with a median age of 53 years.

Thrombotic events are relatively frequent in patients with nephrotic syndrome, being eight times more frequent than in the general population [16]. The hypercoagulable state due to nephrotic syndrome is not well understood and is probably multifactorial [17]. The most frequent reported abnormalities include reduced levels of natural anticoagulants such as antithrombin III, plasminogen, and protein C and S (due to urinary losses), increased platelet activation, hyperfibrinogenemia, inhibition of plasminogen activation, and the presence of high-molecular-weight fibrinogen in the serum (due to increased liver synthesis) [17,18,19,20]. Moreover, hypovolemia, diuretic therapy, and corticosteroid treatment can additionally increase the thrombotic risk.

Comparable to nephrotic syndrome, other hyperlipidemic disorders such as familial hypercholesterolemia are associated with a high incidence of thrombotic episodes [21]. In this case, the level of oxidized low-density lipoproteins (oxLDLs) is increased, and oxLDLs seem to interact with monocytes and macrophages, leading to the expression of tissue factor, a procoagulant molecule [22].

Statins are known to have pleiotropic effects on coagulation and inflammation: improvement in endothelial function, inhibition of platelet activation and of thrombosis [23]. Moreover, studies in non-renal patients reported antithrombotic properties of statins due to their influence on the coagulation cascade [24,25].

In line with this, Zou et al. reported that statin therapy was associated with a lower risk of venous thromboembolism in patients with primary membranous nephropathy, a benefit independent of the statin potency [26]. Similarly, in our cohort of nephrotic patients—more than half with primary membranous nephropathy—statins were associated with a lower risk of thrombotic events.

We acknowledge several limitations of our study. The data were retrospectively collected and were, therefore, dependent on the accuracy and completeness of the electronic databases (for example, smoking status was not available for all the patients, and therefore was not included as a variable in the analysis). In addition, the follow-up time of 24 months was a compromise between the time with data completeness and the clinically relevant time for the investigated outcomes. Importantly, LDL cholesterol, HDL cholesterol, and lipoprotein (a) were not routinely measured and were not included in the analysis. Moreover, asymptomatic thrombotic events may have been missed because the participants were not regularly screened, which could have led to an underestimation of the incidence. Our cohort may not be representative of all adult patients with nephrotic syndrome due to the referral-based nature of our cohort.

## 5. Conclusions

In conclusion, statins did not improve the remission rate and did not reduce the risk of either major cardiovascular events or of kidney replacement therapy initiation in non-diabetic nephrotic patients. However, statins seemed to reduce the risk for thrombotic events. Further randomized controlled studies are needed to establish the role of statin therapy in nephrotic syndrome management.

## Figures and Tables

**Figure 1 medicina-59-00512-f001:**
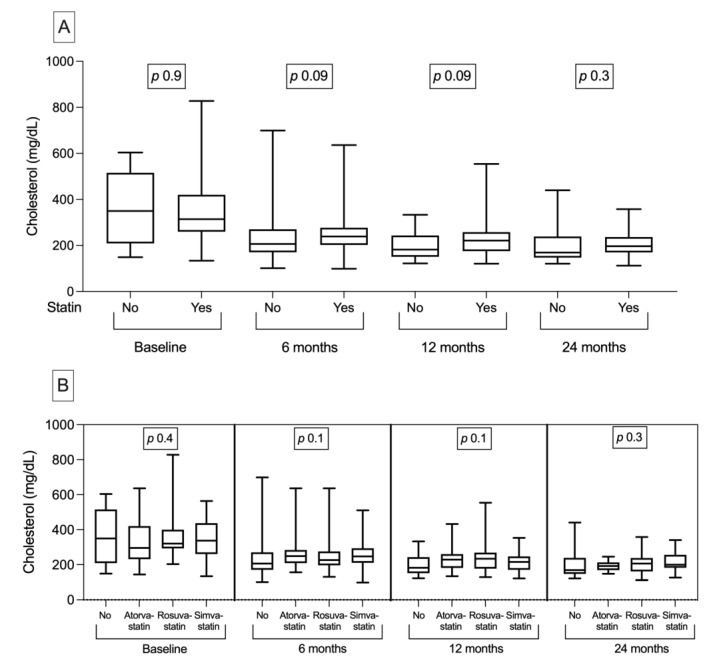
Cholesterol levels according to statin therapy at baseline, 6 months, 12 months, and 24 months; (**A**) statin versus no statin therapy, (**B**) types of statins used versu no statin.

**Figure 2 medicina-59-00512-f002:**
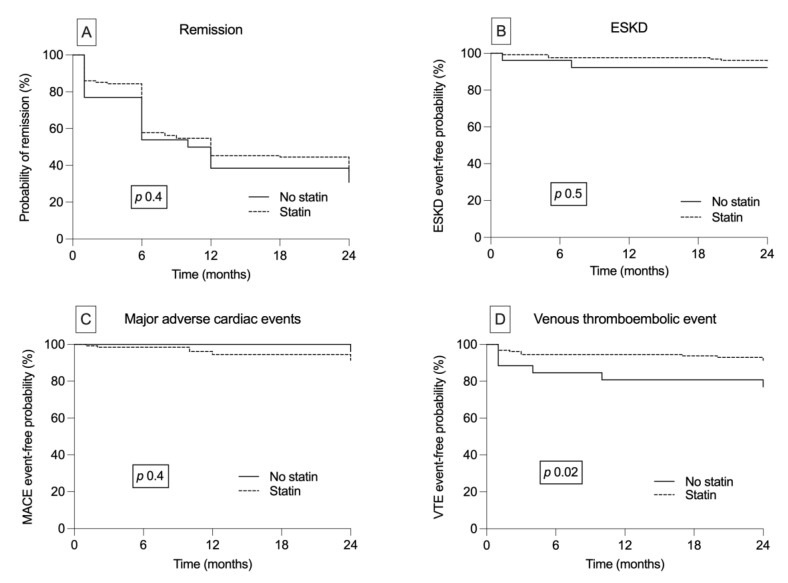
Cumulative probability of nephrotic syndrome remission (**A**) and of event-free survival: (**B**) End-stage kidney disease (ESKD), (**C**) Major cardiovascular events (MACE), (**D**) Venous thromboembolic events (VTE)—according to statin therapy (Kaplan—Meier analysis).

**Table 1 medicina-59-00512-t001:** Patients’ characteristics at baseline and according to statin therapy.

	All (N = 154)	Treatment with Statins		*p*
		Yes (n = 128)	No (n = 26)	
Age (years)	53 [39–64]	53 [40–63]	53 [36–65]	0.6
Male sex (%)	64	64	65	0.8
BMI (Kg/m^2^)	27.0 [25.6–29.0]	27.0 [26.0–29.2]	27.0 [24.7–28.5]	0.3
Nephrotic syndrome cause (%)				0.01
Membranous nephropathy	55	59	39	
Minimal change disease	31	28	42	
FSGS	11	12	8	
MPGN	3	2	12	
Arterial hypertension (%)	44	44	46	0.8
Charlson score	1 [0–2]	1 [0–2]	1 [0–2]	0.6
eGFR (mL/min)	61.9 [45.2–81.0]	61.9 [46.1–81.3]	61.6 [41.0–80.0]	0.7
Serum albumin (g/dL)	2.9 [2.5–3.4]	3.0 [2.6–3.3]	2.8 [2.4–3.0]	0.07
Proteinuria (g/g)	6.4 [4.4–9.0]	6.5 [4.3–9.1]	6.2 [4.6–8.1]	0.7
Hematuria (RBC/mm^3^)	34 [5–75]	30 [5–73]	38 [5–125]	0.7
Hemoglobin (g/dL)	13.4 [11.9–14.8]	13.7 [12.2–14.8]	12.6 [10.1–15.1]	0.1
C-reactive protein (mg/L)	2 [1–6]	2 [1–6]	2 [0–7]	0.5
Cholesterol (mg/dL)	319 [255–420]	314 [260–420]	350 [213–511]	0.9
Triglycerides (mg/dL)	215 [140–305]	215 [140–301]	203 [150–344]	0.7
Total lipids (mg/dL)	1009 [881–1341]	1009 [888–1326]	1015 [713–1448]	0.9
**Treatment**				
Immunosuppression (%)	100	100	100	-
Type of immunosuppression (%)				
Corticotherapy only	26	24	38	0.1
Cyclophosphamide	58	61	43	0.07
Cyclosporine	16	15	19	0.6
RAAS blockade (%)	60	60	62	0.8
**Outcome**				
Response to therapy (%)				0.4
No remission	21	22	12	
Partial remission	15	15	19	
Complete remission	64	63	69	
KRT initiation (%)	5	5	8	0.5
Major cardiovascular event (%)	8	9	4	0.4
Thrombotic complications (%)	11	9	23	0.03

BMI, body mass index; eGFR, estimated glomerular filtration rate; FSGS, focal and segmental glomerulosclerosis; MPGN, membranoproliferative glomerulonephritis; RAAS, renin angiotensin aldosterone system; KRT, kidney replacement therapy.

**Table 2 medicina-59-00512-t002:** Association between statin therapy and remission, ESKD, MACE, and thrombotic complications in four models of Cox regression analysis adjusted for eGFR, proteinuria, and serum albumin at baseline.

Endpoint (Model)	Variables	HR (95%CI)	*p*
Remission	eGFR (mL/min) Proteinuria (24 h) Serum albumin (g/dL) Statin versus no statin therapy	1.00 (1.00, 1.01) 0.94 (0.89, 0.99) 1.16 (0.79, 1.69) 1.17 (0.70, 1.96)	0.02 0.03 0.4 0.5
ESKD	eGFR (mL/min) Proteinuria (24 h) Serum albumin (g/dL) Statin versus no statin therapy	0.96 (0.93, 0.99) 1.03 (0.88, 1.22) 1.23 (0.33, 4.59) 1.57 (0.30, 8.10)	0.03 0.6 0.7 0.5
MACE	eGFR (mL/min) Proteinuria (24 h) Serum albumin (g/dL) Statin versus no statin therapy	0.98 (0.95, 1.00) 0.77 (0.59, 0.98) 3.94 (1.20, 12.87) 0.44 (0.05, 3.59)	0.09 0.04 0.02 0.4
Thrombotic complication	eGFR (mL/min) Proteinuria (24 h) Serum albumin (g/dL) Statin versus no statin therapy	0.99 (0.97, 1.00) 1.10 (1.00, 1.22) 0.67 (0.26, 1.75) 2.83 (1.02, 7.84)	0.3 0.04 0.4 0.04

CI, confidence interval; eGFR, estimated glomerular filtration rate; ESKD, end-stage kidney disease; HR, hazard ratio; MACE, major adverse cardiovascular event.

## Data Availability

The datasets used and/or analyzed during the current study are available from the corresponding author on reasonable request.

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
