# Peer review of "Nephrotic Syndrome and Statin Therapy: An Outcome Analysis"

_medicina, 2023, doi:10.3390/medicina59030512_

Round 1
Reviewer 1 Report
Dear Authors, congratulations for the manuscript. I have just two comments about the manuscript: 1- In de abstract (line 18-18) ..."The mean time to VTE was shorter in the statin group (22.6 18 (95%CI 21.7, 23.6) versus 20.0 (95%CI 16.5, 23.5) months, p 0.02)."...I think you would like to say the opposite. 2- I suggest that you explain in methods what Charlson index is.
Author Response
#Reviewer 1
Dear Authors, congratulations for the manuscript. I have just two comments about the manuscript:
Thank you so much for your kind and positive feedback; we modified our manuscript as the reviewer suggeseted.
1- In de abstract (line 18-18) ..."The mean time to VTE was shorter in the statin group (22.6 18 (95%CI 21.7, 23.6) versus 20.0 (95%CI 16.5, 23.5) months, p 0.02)."...I think you would like to say the opposite.
We modified the text: "The mean time to VTE was longer in the statin group.."
2- I suggest that you explain in methods what Charlson index is.
As the reviewer suggested we modified the methods by including the methodology of the Charlson comorbidity score.
"The Charlson comorbidity score (available at: https://www.mdcalc.com/calc/3917/charlson-comorbidity-index-cci) was used to assess the burden of comorbidities in the studied patients, it is based on a weighted sum of the presence and severity of 17 various conditions."
Reviewer 2 Report
The authors clearly pointed out the importance of more knowledge concerning statin therapy in nephrotic syndrome. Nevertheless, some point are unclear and should be improved.
1. The smoking status of the patients would be of interest and statistical subanalysis
2. Was Lipoprotein a measured as well? Is there a correlation concerning thrombosis?
3. A subanalysis between the age group is missing
4. Is there a difference in the type of station concerning delay in CKD progression in CKD stage 3B-5 patients?
5. How do you explain that there is no difference between the statin types (rosuvastatin with either atorvastatin or simvastatin) on lipid levels? What are the LDL levels?
Author Response
#Reviewer 2
The authors clearly pointed out the importance of more knowledge concerning statin therapy in nephrotic syndrome. Nevertheless, some points are unclear and should be improved.
Thank you for recognizing the importance of our manuscript, we improved our manuscript according to the reviewer suggestions.
- The smoking status of the patients would be of interest and statistical subanalysis
Unfortunately, due to the retrospective design of the study the information regarding the smoking status of the patients was not available for all the included subjects. However, the smoking status was available for 60% of the studied patients; in univariate analysis we found no differences between the two studied groups (Statin therapy 32% vs No statin therapy 34%, p 0.9). We included this as a limit ofour study:
"We acknowledge several limitations to our study. Data were retrospectively collected and were, therefore, dependent on the accuracy and completeness of the electronic databases (for example, smoking status was not available for all the patients, and therefore was not inclued as a variable in the analysis)."
While we acknowledge the importance of smoking status as an important risk factor for the studied outcomes, we tried to characterize the burden of comorbidities (i.e. risk factor) important for the outcome analysis using the Charlson comorbidity score. Moreover, the strength of our study resides on the impact of the statin therapy on the four hard endpoints: ESKD, MACE, thrombotic complications and remission. To the best of our knowledge, this is the first study to include this spectrum of outcome in patients with nephrotic syndrome and statin therapy.
- Was Lipoprotein a measured as well? Is there a correlation concerning thrombosis?
Unfortunately, we did not measure lipoprotein (a) levels in our study, so we are unable to comment on any potential correlation between lipoprotein (a) and thrombosis. We do acknowledge that the lack of measurement of lipoprotein (a) is a limitation of our study and may have an impact on the conclusions that can be drawn from our findings. However, we believe that our study still provides valuable insights into the relationship between statin therapy and thrombosis in nephrotic syndrome patients. The following changes were performed in the manuscript:
"Importantly, LDL cholesterol, HDL cholesterol and lipoprotein (a) were not routinely measured and were not included in analysis."
- A subanalysis between the age group is missing
While we appreciate your suggestion to conduct a subanalysis by age group, we did not find any significant differences in age between the groups we studied (Table 1: Statin therapy 53 (40-63) vs No statin therapy 53 (36-65) years, p 0.6). In addition, our univariate Cox regression analysis did not identify age as a significant predictor of the outcomes we studied.We agree that age is an important factor that could impact our findings, but our relatively young study population (median age 53 years old) may explain the lack of significance in this regard. Nonetheless, we acknowledge that subanalysis by age should be considered for future research in a cohort that includes older patients with nephrotic syndrome.
- Is there a difference in the type of statin concerning delay in CKD progression in CKD stage 3B-5 patients?
The number of patients with CKD 3B-5 in our cohort was rather small - 37 patients; therefore, an analysis of kindey survival from the perspective of the type of statin is not possible due to the limited cohort and the reduced number of events. However, we performed a Kaplan-Meier analysis on the type of statin used for the kidney survival on the enteire cohort; we found no difference between the type of statin used and the time to ESKD. The following changes were performed in the manuscript:
" Moreover, there were no differences in kidney survival between the types of statins used (Kaplan-Meier analysis, log rank test, p 0.1)."
- How do you explain that there is no difference between the statin types (rosuvastatin with either atorvastatin or simvastatin) on lipid levels? What are the LDL levels?
In our study statin therapy was not associated with a significant impact on the lipid levels and did not reduce the risk of MACE probably due to the pathogenic treatment (i.e. immunosupression) which lead to the remission of the nephrotic syndrome in almost 80% of the studied patients.
Statin therapy should be considered in the treatment of resistant nephrotic syndrome when the secondary hyperlipidemia is likely to persist. This treatment resistant group is more likely to develop ESKD, arterial hypertension, atherosclerosis and to increase the cardiovascular burden.
This has been explained in the discussion section of the manuscript:
" The statin effect on cholesterol level and on MACE could be confounded by the pathogenic treatment (i.e. immunosuppressive treatment) which leads to the remission of the nephrotic syndrome."
"Nevertheless, persistent nephrotic syndrome and hyperlipidemia seem to be risk factors for atherosclerotic cardiovascular disease, especially if other cardiovascular risk factors are present. However, the low incidence of MACE in our cohort could be explained by the rather mild Charlson score and the relatively short follow-up time for a population with a median age of 53 years old."
We agree with the reviewer that the absence of the HDL and LDL cholesterol values is an important limitation of our study; we highlighted this as an important limitation of our study. Nevertheless, the strength of our study resides on the impact of the statin therapy on the four hard endpoints: ESKD, MACE, thrombotic complications and remission. To the best of our knowledge, this is the first study to include this spectrum of outcome in patients with nephrotic syndrome and statin therapy.
Round 2
Reviewer 2 Report
I would accept the manuscript.